# Low Molecular Weight Hyaluronic Acid Effect on Dental Pulp Stem Cells In Vitro

**DOI:** 10.3390/biom11010022

**Published:** 2020-12-28

**Authors:** Jan Schmidt, Nela Pilbauerova, Tomas Soukup, Tereza Suchankova-Kleplova, Jakub Suchanek

**Affiliations:** 1Department of Dentistry, Charles University, Faculty of Medicine in Hradec Kralove and University Hospital, 500 05 Hradec Kralove, Czech Republic; Jan.Schmidt@lfhk.cuni.cz (J.S.); Tereza.Suchankova@lfhk.cuni.cz (T.S.-K.); SuchanekJ@lfhk.cuni.cz (J.S.); 2Department of Histology and Embryology, Charles University, Faculty of Medicine in Hradec Kralove, 500 03 Hradec Králové, Czech Republic; SoukupTo@lfhk.cuni.cz

**Keywords:** hyaluronic acid, dental pulp stem cells, low molecular weight hyaluronic acid, tissue engineering, scaffold

## Abstract

Hyaluronic acid (HA) and dental pulp stem cells (DPSCs) are attractive research topics, and their combined use in the field of tissue engineering seems to be very promising. HA is a natural extracellular biopolymer found in various tissues, including dental pulp, and due to its biocompatibility and biodegradability, it is also a suitable scaffold material. However, low molecular weight (LMW) fragments, produced by enzymatic cleavage of HA, have different bioactive properties to high molecular weight (HMW) HA. Thus, the impact of HA must be assessed separately for each molecular weight fraction. In this study, we present the effect of three LMW-HA fragments (800, 1600, and 15,000 Da) on DPSCs in vitro. Discrete biological parameters such as DPSC viability, morphology, and cell surface marker expression were determined. Following treatment with LMW-HA, DPSCs initially presented with an acute reduction in proliferation (*p* < 0.0016) and soon recovered in subsequent passages. They displayed significant size reduction (*p* = 0.0078, *p* = 0.0019, *p* = 0.0098) while maintaining high expression of DPSC markers (CD29, CD44, CD73, CD90). However, in contrast to controls, a significant phenotypic shift (*p* < 0.05; CD29, CD34, CD90, CD106, CD117, CD146, CD166) of surface markers was observed. These findings provide a basis for further detailed investigations and present a strong argument for the importance of HA scaffold degradation kinetics analysis.

## 1. Introduction

Hyaluronic acid (HA) is an acidic, non-sulfated glycosaminoglycan with a repeating disaccharide structure of d-glucuronic acid and N-acetyl-d-glucosamine. From these simple dimers, long linear polymer chains counting thousands of repetitions and a molecular weight ranging up to 10 MDa are formed [1]. As a ubiquitous component of bacterial, fungal, and animal extracellular matrix with biocompatible and biodegradable properties, HA is considered a promising material for tissue engineering. HA is also widely distributed in the human body and can be found in the umbilical cord, synovial fluid, dental pulp, vitreum, or epithelial and connective tissues [2,3].

The general biological functions of HA are hydration, space-filling capacity, lubrication, and forming of the framework through which cells migrate [4]. While its main binding receptors are cluster determinant (CD) 44, receptor for hyaluronate-mediated motility (RHAMM), lymphatic vessel endothelial hyaluronan receptor-1 (LYVE-1), and intercellular adhesion molecule 1 (ICAM-1), HA affects cell motility, adhesion, proliferation, and differentiation [5,6].

In the human body, HA is synthesized and predominantly present at a high molecular weight (HMW-HA, MW > 1000 kDa), and thus, long-chain molecules of HA are a common part of the natural environment of cells [7]. HMW-HA has anti-inflammatory, anti-angiogenic, and anti-tumor properties, and supports tissue integrity [4,8]. During the degradation of HA, its molecules are cleaved into smaller fragments of low molecular weight (LMW-HA, MW < 500 kDa) by a family of enzymes called hyaluronidases (Hyals) or via non-enzymatic reactions involving alkaline, acidic hydrolysis, and oxidant decomposition [9]. Two major Hyals are Hyal-1 and Hyal-2. Hyal-2 degrades HA into fragments of approximately MW 20 kDa, Hyal-1 cleaves HA up to tetrasaccharides with an approximate MW of 1600 Da [10]. Subsequent digestion is performed in lysosomes after the fragments are endocytosed [4]. Under physiological conditions, the HA turnover is constant [7]. However, this balance may be impaired by a breach of tissue integrity or pathological processes such as pH, temperature, thermal, mechanical, free radical, ultrasonic, and enzymatic stresses, which accelerate HA degradation, and hence, increase the concentration of LMW-HA [8,11].

HA plays an essential role throughout the whole process of wound healing. In the initial phase, extravasation of blood leads to the interaction of HA with fibrinogen that commences the extrinsic clotting pathway [11]. The platelets accelerate HA synthesis, and as the framework of long molecules gets denser, it binds more water and saturates the environment with fluid, which clinically manifests as oedema [12]. Such a condition supports cell migration, including chemotactic attraction of inflammatory cells that trigger the inflammatory reaction. During the further phases of wound healing, native HA and its fragments, resulting from degradation processes, modulate the inflammatory response, ingrowth of endothelial cells, organization of granulation tissue, and reepithelization [4,11,13,14,15].

It is important to note each fraction of HA has its essential role accentuated or abated based on the current state of the environment. However, increasing evidence indicates that the impairment of the endogenously regulated mechanism of HA synthesis, cleavage, or degradation may cause or contribute to pathological processes, such as cancer initiation, progression, metastasis, and therapy resistance [16,17,18,19]. While much remains to be explored, there is enough evidence to consider the bioactivity of HA size-dependent, and as such, it needs to be assessed separately for each fraction [20,21].

Small fragments of HA have been proved as potent pro-inflammatory and pro-angiogenic molecules and play a crucial role in cancer progression [4,16]. LMW-HA has also been reported to have a different impact on fibrocyte differentiation and the phenotype of macrophages. While HMW-HA induces differentiation of monocytes into fibrocytes, LMW-HA inhibits fibrocyte differentiation [22]. HMW-HA promotes alternative macrophage activation associated with pro-resolving gene transcription, while LMW-HA encourages them to produce pro-inflammatory mediators associated with the classically activated state [23].

In tissue engineering, HA is considered a promising material thanks to its biocompatible, biodegradable, and bioresorbable properties, as well as its chemical properties and high level of technical processing [24,25]. It is one of the main components of the extracellular matrix, and in high molecular form, it is a part of the cell native habitat. Based on the processing method, HA can be produced in many forms and shapes, opening a wide range of applications, e.g., gels of various viscosity, porous, sponge-like scaffolds, solid prefabricates, or nanomaterials [25,26,27,28]. Chemical and bioactive traits of HA, such as the ability to bind water molecules or forming a framework for cell migration, provide a convenient environment for the seeded cells, while its conductive function facilitates ingrowth of the surrounding tissues [25,29]. The application of HA is broad, including bone, cartilage, skin, eye, periodontal, or dental pulp tissue engineering [30,31,32,33,34].

The optimal rate of degradation occurs when scaffold is gradually replaced by regenerated tissue [24,35]. The degradation rate of HA may be adapted by chemical modifications, such as esterification, crosslinking, or oxidation. However, the degradation process is affected not only by the material properties but also by many various factors resulting from the adjacent tissue [36]. A methodology for controlling degradation kinetics is still a challenging issue with no satisfactory resolution [25,35,36,37]. Additionally, excessive degradation rate exceeding tissue regeneration not only negatively affects the primary supportive function of implanted material but can also lead to an accumulation of scaffold components, which can affect seeded cells and the surrounding environment [38,39]. As the HA degradation products are biologically very potent molecules, this provides a strong argument for assessing their impact on cells in contact with HA materials.

One of the cell types related to HA materials for tissue engineering are dental pulp stem cells (DPSCs). Based on their multilineage differentiation potential, high proliferation activity, and self-renewal, DPSCs are considered an auspicious mesenchymal cell population for cell-based therapy and tissue regeneration [40,41]. Because of the non-invasive access and simple isolation, the dental pulp is an attractive alternative source of mesenchymal stem cells [41]. DPSCs can be obtained from an extracted tooth, most often third lower molars, as part of planned treatment. After extraction, these teeth are considered biological waste and discarded. Hence, isolation of the dental pulp does not represent an intervention for the patient beyond the usual treatment, which would be performed only for the purpose of cell isolation [42]. The capability for differentiating into various cell types, such as odontoblasts, osteoblasts, chondroblasts, endothelial cells, and neural cells, indicates their possible application in the fields of regenerative medicine and tissue engineering [43,44,45].

The dental pulp is located in a cavity inside the tooth, also called the pulpal chamber, and in the root canals. The only connection of this otherwise hermetically sealed zone to the external environment is the physiological foramen of the root apex with a diameter between 0.2 and 0.5 mm [46]. The inner space of the tooth maintains a specific microenvironment separated from the surrounding tissues, and the stem cells housed in the dental pulp are thus protected from differentiation stimuli and remain in the naive form. One of the components of the dental pulp extracellular matrix is HA [47]. As a part of the pulpal niche, HA is assumed to be applicable material for DPSCs scaffolds.

Several findings identify HA as non-toxic material that does not negatively affect viability, proliferation activity, or differentiation potential of DPSCs [48,49,50]. Additional authors present gene expression changes in DPSCs after being exposed to HA, while only a few reports investigate the effect of HA on DPSC surface markers [48,51,52,53,54,55,56]. However, these studies do not assess the biological properties of discrete HA fractions of different sizes and are often confounded by the addition of additives that may affect cell properties. Furthermore, the impact of LMW-HA on DPSCs has not been a focus of these studies.

The aim of this work, therefore, is to evaluate the effect of pure and defined molecular fractions of LMW-HA, which correspond to Hyal-1 and Hyal-2 cleavage products, on DPSC viability, proliferation activity, and surface markers expression after long term cultivation in vitro.

## 2. Materials and Methods

The isolation protocol and the methodology of the experiment were approved by the Ethics Committee of the University Hospital Hradec Kralove (approval number: 201812 S07P, approved: 08 November 2018). Tooth donors or their legal representatives were informed about the content of the experiment before the extraction and signed an informed consent with the use of DPSCs for research purposes.

### 2.1. Hyaluronic Acid

Hyaluronan-derived oligosaccharides of molecular weights 800 Da, 1600 Da, and 15,000 Da used in this study were kindly provided by Contipro a.s. (Dolni Dobrouc, Czech Republic). Hyaluronan oligosaccharides were produced by partial digestion of high-molar mass hyaluronan of biotechnological origin (non-animal, non-genetically modified organisms) with leech hyaluronidase and chromatographic fractionation into size-uniform HA oligosaccharides by anion-exchange chromatography after the enzyme removal. The pharmaceutical grade purity and size of each HA oligomer were confirmed by high-performance liquid chromatography analyses and mass spectrometry. HA oligomers used in this study are declared soluble in aqueous media.

### 2.2. Tooth Collection

Tooth extraction was performed under standard conditions under local anaesthesia. The sex and age of the donors, as well as the teeth specification, are listed in Table 1. Each extracted tooth was wiped with sterile gauze to remove the microbial plaque from to surface and decontaminated in a 0.2% solution of chlorhexidine gluconate for 30 s. Each tooth was transferred in a tube containing transportation medium composed of water for injection (Bieffe Medital, Grosotto, Italy), 10% Hank’s balanced salt solution (HBSS) (Invitrogen, Waltham, MA, USA), 200 μg/mL gentamicin (Invitrogen), 200 U/mL penicillin (Invitrogen), 200 μg/mL streptomycin (Invitrogen), and 1.25 μg/mL amphotericin (Sigma-Aldrich, St. Louis, MO, USA). Fully immersed in the transportation medium, the teeth were transported to the tissue culture laboratory in a cooling box at 4 °C. The period between the extraction and the pulp isolation was <4 h.

### 2.3. Dental Pulp Isolation

The teeth were removed from the transport medium under sterile conditions in a laminar box. The dental pulp of the teeth with incomplete root development was extracted through open apical foramina using a probe and tweezers. The dental pulp of the teeth with completed root development was extracted with a sharp excavator from the dental pulp cavity accessed after crown separation with Luer forceps.

### 2.4. Cell Mobilisation, Expansion, and Banking

Acquired soft tissue was minced by scissors and homogenized in a mini tissue grinder (Radnoti, Covina, CA, USA) with an isotonic solution (phosphate-buffered saline, PBS, (Sigma-Aldrich, St. Louis, MO, USA)). The tissue homogenate was transferred into a tube and enzymatically digested using 0.05% trypsin (Gibco, Thermo Fisher Scientific, Foster City, CA, USA) for 10 min at 37 °C. After centrifugation (2000 rpm, 5 min), the pellet was resuspended in an adherent tissue culture dish (TPP, Sigma-Aldrich) with a standard cultivation medium for mesenchymal adult progenitor cells containing Minimum Essential Medium (MEM Alpha Medium, Gibco, Thermo Fisher Scientific, Foster City, CA, USA) and 2% fetal bovine serum (FBS, PAA Laboratories, Dartmouth, MA, USA), supplemented with 1% Insulin-Transferrin-Selenium (Invitrogen) 10 ng/mL epidermal growth factor (PeproTech, London, UK), 10 ng/mL platelet-derived growth factor (PeproTech), 50 mM dexamethasone (Bieffe Medital), 0.2 mM L-ascorbic acid (Bieffe Medital), 2% glutamine (Invitrogen), 100 U/mL penicillin, 100 µg/mL streptomycin (Invitrogen), 20 μg/mL gentamicin (Invitrogen), and 2.5 µg/mL amphotericin (Sigma-Aldrich). The culture dish with culture medium and seeded cells was incubated at 37 °C in a humidified atmosphere containing 5% CO_2_. The standard culture medium was changed every 3 days, and, prior to this, the tissue culture dish was washed by PBS to remove non-adherent elements and detritus. After reaching 70% confluence, the tissue culture dish was washed by PBS, DPSCs were passaged using 0.05% trypsin-EDTA (Gibco) and reseeded in a concentration of 5000 cells per cm^2^ in a new culture dish. After they expanded, the cell lines were cryopreserved using an uncontrolled freezing rate technique with 10% dimethyl sulfoxide as a cryoprotective agent and stored in a cell bank at –80 °C.

### 2.5. Cell Cultivation

During the experiment, cell lines 1–5 were thawed in a 37 °C thermal bath and seeded in a standard culture medium. After expanded, the cells were harvested, counted, and seeded in experimental media in a concentration of 5000 cells per cm^2^ in parallel. The experimental media differed from the standard culture medium only in the enrichment of 100 μmol/L of pharmaceutical grade HA (Contipro a.s., Dolni Dobrouc, Czech Republic) dissolved in the culture medium, each of a different low molecular weight, as follows: experimental medium 1 (E1) 800 Da HA, experimental medium 2 (E2) 1600 Da HA, experimental medium 3 (E3) 15,000 Da HA. For control data, cell lines 1–5 cultivated in the standard culture medium (control) were used. All cells were cultivated under the same controlled condition for 6 passages until they exceeded the Hayflick limit. For clearer data presentation, passage 0 refers to the passage after thawing, while passages 1–6 refer to the passages of the experiment. The protocol described above was followed throughout the cultivation.

### 2.6. Cell Characteristics

The cell count and cell diameter were measured using Z2-Counter (Beckman Coulter, Miami, FL, USA) in every passage. Proliferation activity was determined in each passage as population doublings (PDs). To calculate PD reached in every passage, the following formula was used:PD = log2(N*x*/N1)

N*x* is the total passage cell count, and N1 is the initial cell count seeded into the culture dish (5000 cells/cm^2^). At the end of the cultivation, cell viability was assessed based on the trypan dye exclusion method using the Vi-Cell analyzer (Beckman Coulter), and cell surface markers phenotype analysis was performed using flow cytometer Cell Lab Quanta (Beckman Coulter). DPSCs were detached using the 0.05% trypsin-EDTA solution (Gibco) and stained with phycoerythrin (PE) or fluorescein isothiocyanate (FITC) labeled immunofluorescence antibodies against cluster of differentiation markers (CD): CD29 (TS2/16, BioLegend, San Diego, CA, USA), CD31 (MBC 78.2, Invitrogen), CD34 (581 (Class 287 III), Invitrogen), CD44 (MEM 85, Invitrogen), CD45 (HI30, Invitrogen), CD49f (GoH3, Invitrogen), CD73 (AD2, BD Biosciences 288 Pharmingen, Erembodegen, Belgium), CD90 (F15-42-1-5, Beckman Coulter, USA), CD105 (SN6, 289 Invitrogen), CD106 (STA, BioLegend, USA), CD117 (2B8, Chemicon, Tokyo, Japan), CD133 (13A4, eBioscience), CD146 (TEA1/34, Beckman Coulter), CD166 (3A6, Beckman Coulter), CD271 (ME20.4, BioLegend). All analyses were performed according to the manufacturer’s instructions.

### 2.7. Statistical Analysis

For statistical analysis, Friedman’s test followed by Dunn’s post hoc test was performed for non-parametrically distributed data. One-way ANOVA followed by Fisher’s LSD post hoc test were performed for parametrically distributed data. For data sphericity determination, Mauchly’s sphericity test was performed. Statistical significance was determined as *p* < 0.05 and labeled as follows: * *p* < 0.05, ** *p* < 0.005, *** *p* < 0.001. GraphPad Prism (version 8.0.0 for Windows, GraphPad Software, San Diego, CA, USA) and IBM SPSS Statistics (IBM Corp, Released 2017. IBM SPSS Statistics for Windows, Version 25.0. Armonk, NY, USA: IBM Corp.) were used for all calculations.

## 3. Results

The data presented are for the cell populations cultivated in control/E1/E2/E3 media, respectively, and were measured at the end of the experiment, unless otherwise stated.

### 3.1. Cell Viability

High DPSC viability was maintained across all groups, and no negative impact of LMW-HA on cell viability was observed. All experimental groups consistently showed viability greater than 94% in contrast to the control group, the viability of which ranged from 80.2 to 97.2%. Compared to control, the viability of experimental groups was higher in three lines (lines 1–3) and similar in two lines (lines 4 and 5).

The viability of the DPSCs in order control/E1/E2/E3 was as follows: Line 1: 89.0/96.9/97.1/98.0%; line 2: 80.2/97.4/97.2/96.6%; line 3: 90.3/96.5/98.0/98.3%; line 4: 96.4/96.4/98.7/96.2%; line 5: 97.2/93.9/94.8/96.2%, as illustrated in Figure 1.

### 3.2. Cell Size

The median value of the cell diameter was determined for each line, and statistical analysis was performed. DPSCs cultivated in E1, E2, and E3 were significantly smaller in diameter compared to those cultivated in the control medium. Control-E1: *p* = 0.0078, control-E2: *p* = 0.0019, control-E3: *p* = 0.0098. Among the experimental groups, no significant difference in cell size was observed.

The median of the cell diameter was: Line 1: 13.5/9.7/9.7/12.0 μm; line 2: 10.3/10.3/10.4/10.8 μm; line 3: 15.9/11.4/11.1/11.1 μm; line 4: 15.8/14.7/14.6/14.3 μm; line 5: 11.8/9.7/9.9/10.2 μm, as illustrated in Figure 2.

### 3.3. Proliferation Activity

Cumulative population doublings (cPDs) achieved in two of three experimental groups were statistically different compared to the control group (Figure 3). Control-E1: *p* = 0.0015, control-E2: *p* = 0.0143, control-E3: *p* = 0.0864. Among the experimental groups, no significant difference in cPDs was observed. A detailed assessment of the proliferative activity in every single passage revealed a decrease in experimental groups (*p* < 0.0016) in the first passage (Figure 4). This passage was the passage HA was firstly added to the cultivation media.

Total population doublings achieved: Line 1: 23.2/22.1/21.2/22.8; line 2: 20.3/16.8/17.2/16.3; line 3: 22.4/21.0/21.3/21.6; line 4: 24.0/18.1/18.9/19.1; line 5: 25.6/18.0/18.8/19.2.

Individual population doublings data are listed in Appendix A.

### 3.4. Surface Markers Analysis

Except for CD133 and CD271, the control group showed higher positivity for examined surface markers compared to experimental groups: CD29 (*), CD31, CD34 (*), CD44, CD45, CD49f, CD73, CD90 (***), CD105, CD106 (*), CD117 (*), CD146 (*), CD166 (***). For CD271, the positivity of cells was approximately the same, and for CD133, experimental groups expressed higher positivity than the control group. Detailed statistical analysis is illustrated in Figure 5.

## 4. Discussion

HA and DPSC are attractive research topics, and their combined use in the field of tissue engineering seems to be very promising. However, a profound characterization of the HA effect on DPSC is still required. As a biodegradable material with bioactive functions dependent on the size of the molecule, the properties of HA must be evaluated separately for each molecular weight fraction.

The existing literature on HA impact on DPSCs appears to be mostly focused on HMW-HA. The number of works dedicated to LMW-HA is very limited. Additionally, most experiments took place in vitro, and thus, the biodegradation of HA was not taken into account.

In this study, we decided to evaluate the effect of precisely defined and pharmaceutical grade LMW-HA fractions, i.e., 800 Da, 1600 Da, and 15,000 Da, on DPSCs. The purity of HA is important because products of lower than pharmaceutical grade quality may contain endotoxins and other impurities that elicit an inflammatory response that may be incorrectly attributed to HA [57]. The choice of molecular weights was not random but determined by the size of the fragments resulting from Hyals cleavage in vivo. We consider this specification as very relevant because artificial fragments of HA produced in various sizes may be formally considered as LMW (<500 kDa); however, in such a broad range, they may not meet the bioactive properties as the particular cleavage products of Hyals. As far as we know, this is the first work describing the impact of HA fragments of similar MW as the cleavage products of Hyal-1 and Hyal-2 on DPSCs. We designed this study with discrete LWM-HA-sized fractions to assess whether molecular size within a range of 800–15,000 Da may have a different impact on bioactive properties in DPSCs, but after careful analysis, we found that this was not the case. Furthermore, since these LMW fractions all display similar characteristics in our DPSC model, we may consider them an alternative to technical replicates.

We assumed LMW-HA would not have a cytotoxic effect on DPSCs, as HA is part of their natural environment which was shown to be true due to viability analysis. Moreover, in three of five lines, increased viability of experimental groups was observed. However, these results were not statistically significant when compared to the control group. Additionally, a strong contrast in size was shown by cell diameter analysis. DPSCs cultivated in experimental media were smaller than cells cultivated in the control medium. These differences were statistically significant. According to previous findings, mesenchymal stem cells (MSCs) size may be highly associated with senescence in vitro [58]. Thus, these results may lead to a hypothesis that LMW-HA decelerated the progression of DPSCs senescence. Such an assumption is in accordance with Alessio et al., who described the influence of LMW and HMW HA on the replicative senescence of MSCs [59]. This unintended and significant finding opens up an opportunity for further investigation.

The analysis of proliferation activity showed a reduction in the total population doublings of cells cultivated in media enriched with LMW-HA. This disparity was statistically significant in two of the three experimental groups. A detailed analysis of proliferation activity in every single passage revealed an initial rapid and statistically significant decline in experimental groups in the first passage. This phenomenon was solitary and was not observed in subsequent passages. The first passage was the time point in which the cells were initially seeded in LMW-HA enriched culture media. Therefore, we hypothesize the decrease of proliferation activity was caused by this sudden media modification. This finding shall be considered in experiments assessing the impact of LMW-HA on DPSCs with a short time frame as their results may be affected by this reversible phenomenon and thus not correspond with the long-term effects of scaffold application.

As this study is the first to report on the effect of HA fragments in the size of Hyals products on DPSCs, we decided to focus not only on cell surface markers for which HA-related changes have already been reported, but on a broad panel of CDs reflecting stemness, differentiation, motility, or adhesion.

For mesenchymal adult stem cell markers, the expression of CD29, CD44, and CD90 was examined. On average, high positivity was achieved in every group; however, the expression was consistently lower in experimental groups and significantly so for CD29 and CD90. High-level expression of stromal associated markers CD 73 and CD 166 was observed in the control group. In experimental groups, a high expression of CD73 was shown as well. Contrarily, CD166, an adhesion molecule, expressed in MSCs and fibroblasts, was significantly downregulated in experimental groups. A similar disparity was also observed in the expression of CD105 (endoglin), a membrane glycoprotein involved in angiogenesis [60]. This result corresponds with Bringhof et al., who has reported a positive correlation between CD105 and CD166 expression in MSCs [61]. These findings indicate that LMW-HA negatively affects DPSCs in the expression of fibroblast and angiogenesis-related surface markers.

CD 31 and CD106, adhesion molecules typical for endothelial cells, did not exceed low-level expression in any group. However, much lower expression was observed for CD106. While cells in the control group maintained approximately 14% positivity for CD106, its average expression in experimental groups was <1%. CD106, also known as vascular cell adhesion molecule 1 (VCAM-1), is a cell surface protein mediating adhesion of leukocytes and endothelial cells and exhibits cytokine-inducible expression [62]. Yang et al. have reported CD106 to be a possible biomarker for a subpopulation of MSCs with unique immunosuppressive activity [62]. Our results indicate that LMW-HA, which exerts pro-inflammatory properties, significantly downregulates the CD106+ subpopulation of DPSCs.

CD49f (Integrin alpha-6, ITGA6) expression in experimental groups was notably lower compared to the control group. In a recent publication of Nieto-Nicolau et al., it has been demonstrated that MSCs expressing higher levels of CD49f show higher clonogenicity, migration, and differentiation potential and that its expression is affected by culture conditions and extracellular matrix proteins. Their findings have also pointed out that these changes are regulated through the protein kinase B (Akt) pathway and cell cycle inhibitor proteins p53 and p21 [63]. Similarly, we have observed a statistically significant decrease of CD117 (transmembrane receptor tyrosine kinase, c-KIT) expression in experimental groups. CD117 signaling also affects the Akt pathway and plays a role in cell proliferation, differentiation, adhesion, motility, and angiogenesis in hematopoietic cells and cancer stem-like cells [64,65]. Our data show the downregulation of both of these membrane proteins related to proliferation, mobility, and differentiation in DPSCs cultivated in LMW-HA enriched media. Based on these findings we hypothesize LMW-HA may affect proliferation, mobility, and differentiation of DPSCs.

Additionally, biological variability was found between cell lines. While CD133 (prominin-1) expression was uniformly low in the control group, data from experimental groups varied. The differences were not random but followed a solid pattern as they were bound to particular biological replicates. In E1/E2/E3, CD133 positivity in lines 1 and 4 compared to lines 2, 3, and 5 was 5.6-fold, 5.6-fold, and 4.3-fold higher, respectively. As such changes were not observed in the control group, we attribute it to possible interindividual susceptibility of DPSCs to LMW-HA. It is important to note that CD133 expression in DPSCs is not well-defined, differs across studies, and needs to be further investigated [66].

The opposite event was observed in the expression of CD146 (melanoma cell adhesion molecule, MCAM) with consistent levels within all experimental groups while incongruent results in the control group. This surface marker is associated with vascular smooth muscle commitment of MSCs and pericytes [67]. However, in DPSCs, CD146+ subpopulation was also identified as dentin/pulp-like structures promotors with high expression of alkaline phosphatase (ALP), a mineralization marker for odontoblast/osteoblast-like DPSCs differentiation [68,69]. Therefore, the results may be related to the discrete tendency of DPSCs to differentiate into odontoblast/osteoblast-like cells. High variability of CD146 in DPSCs has also been reported by other authors, indicating that there may be subpopulations of DPSCs with different expressions of this surface marker [70,71,72]. Although the difference in CD146 expression between the control and experimental groups of our experiment was statistically significant, we consider it relevant to comment on this result in detail due to the data spread in the control group contrasting with experimental groups. The variableness of DPSCs is intensively studied, and we believe the phenomenon we observed may contribute to this trending research topic [73,74,75]. Controls of line 1 and line 2 were positive for CD146 in 82.5% and 58.9%, respectively. On the other hand, controls of lines 3, 4, and 5 were CD146 positive in 0.2%, 4.8%, and 6.7%, respectively. In contrast, experimental groups cells maintained CD146 positivity in the short-range between 0.02–1.33%. Even though the cause of the various expression of CD146 within the control group has not been revealed, it is apparent that all cells cultured in LMW-HA enriched media, including lines 1 and 2, maintained CD146 expression consistently at very low levels. We can hypothesize whether the distant values were due to the interindividual variability of DPSCs and, if so, whether LMW-HA was responsible for suppressing its manifestation in experimental groups. Additionally, we were interested in assessing the differences between the control group and experimental groups without the distant values of cell lines 1 and 2. Thus, we further focused on cell lines 3–5. The statistical analysis revealing significant differences between control-E1 and nearly significant differences between control-E2 in cell lines 3–5 is included in Appendix A. These findings may indicate that although CD146 expression is variable in DPSCs, LMW-HA leads to its suppression in all DPSCs regardless of their original expression level of this surface marker. However, to confirm such a conclusion, further investigation is required.

In correspondence with the definition of MSCs presented by Dominici et al., we observed low expression of CD34 and CD45 in all groups. CD34, a transmembrane phosphoglycoprotein, is a typical marker for primitive pluripotent stem cells, both stromal and hematopoietic [76]. Alraies et al. reported CD34 as a potential marker for a subset of DPSCs of stromal stem cell population with neural crest origin [77]. Their findings also indicate that the distribution of CD34+ cell subset is different between donors and even between individual teeth within one donor. In conclusion, they suggest CD34 analysis to be a possible tool for the identification of populations rich in this cell subtype to further clinical application. Although CD34+ cell expression was classified as low in all groups of our study, a statistically significant difference between DPSCs cultured in control and experimental media was observed. Further analysis involving cell sorting may reveal in detail the LMW-HA effect on this DPSCs subtype with neural crest commitment.

CD271 (low-affinity nerve growth factor receptor, NGFR, or p75 neurotrophin receptor) has been identified as a marker with affinity to MSCs from diverse sources, including dental-related tissues [78,79]. Alaires et al. demonstrated CD271 expression as a viable marker of high proliferative capacity and multi-potent DPSCs populations. In our study, no significant differences across the groups were observed, and CD271+ positive fraction of DPSCs within the groups corresponded to the findings reported previously by Alvarez [79].

In accordance with the definition of MSCs and DPSCs surface markers, all groups of cells maintained high positivity for CD29, CD73, and CD90, as well as low positivity for CD34 and CD45 [80,81]. However, cells cultivated in LMW-HA enriched media did not meet the criteria for high CD105 expression. The positivity for CD29 and CD90 was significantly higher in cells cultivated in a standard culture medium, and no significant difference was found in CD73 and CD271 expression. Cells cultivated in a standard medium displayed superior expression of surface markers associated with cell mobility, adhesion, and proliferation activity as well as fibroblast, odontoblast, and angiogenesis-related markers compared to cells cultivated in LMW-HA enriched media. Conversely, the presence of LMW-HA elevated the expression of CD133.

## 5. Conclusions

While previous studies assessed the impact of HA on the viability, proliferation, and differentiation of DPSCs, they did not discriminate the bioactive properties of differently sized HA fractions. Our study shows that LMW-HA can impact the proliferation activity, cell size, and surface markers (phenotype) of DPSCs. Further investigation is needed to determine HA scaffold degradation kinetics and its impact on DPSCs.

## Figures and Tables

**Figure 1 biomolecules-11-00022-f001:**
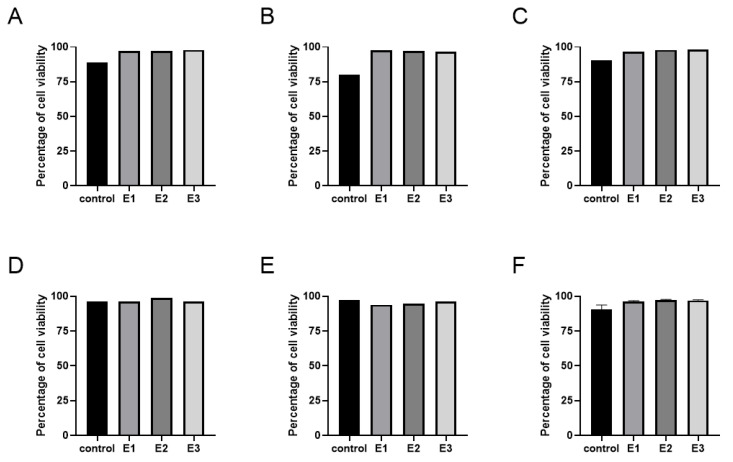
Cell viability at the end of the 6th passage of the experiment. The graphs demonstrate the percentage of living cells in the population of cell line 1 (**A**), cell line 2 (**B**), cell line 3 (**C**), cell line 4 (**D**), cell line 5 (**E**), and as a mean of lines 1–5 with SEM plotted as an error bar (**F**). Cells were cultivated in a standard medium (control) and experimental media 1–3 (E1–3). No statistical significance between the control and experimental groups was found.

**Figure 2 biomolecules-11-00022-f002:**
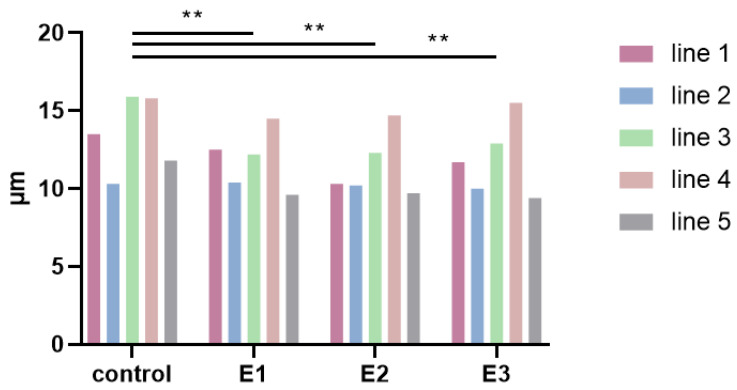
The median of the cell diameter at the end of the 6th passage. The graph demonstrates cell lines 1–5 cultivated in the control (control) and experimental media 1–3 (E1–3). Statistical analysis between pooled data of the control group and pooled data of experimental groups was performed using one-way ANOVA with Fisher’s Least Significant Difference (LSD) post hoc test; ** *p* < 0.005.

**Figure 3 biomolecules-11-00022-f003:**
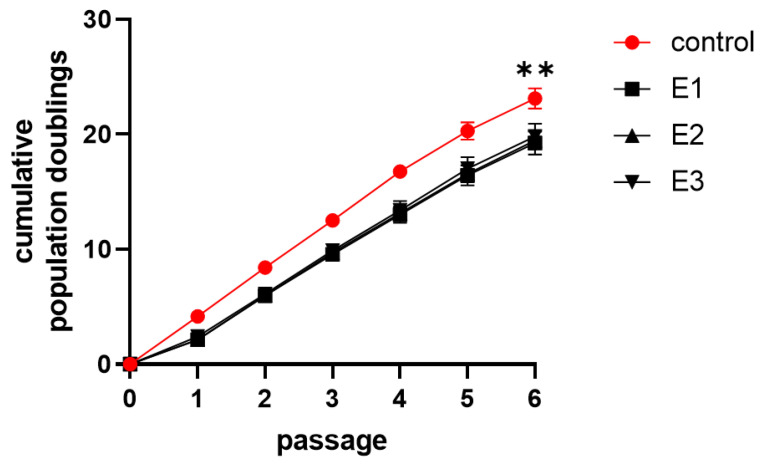
Cumulative population doublings during the cultivation. The graph demonstrates cumulative proliferation doublings of dental pulp stem cells as a mean of cell lines 1–5 with SEM plotted as error bars in experimental passages. Cells were cultivated in the control (control) and experimental media 1–3 (E1–3). Statistical analysis was performed using Friedman’s test followed by Dunn’s post hoc test; ** *p* < 0.005. Individual graphs for each cell line are enclosed in Appendix A.

**Figure 4 biomolecules-11-00022-f004:**
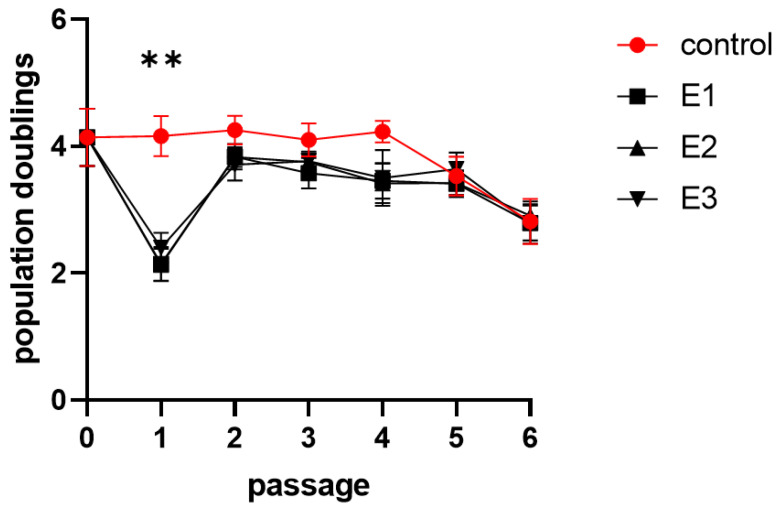
Population doublings during the cultivation. The graph demonstrates population doublings of dental pulp stem cells as a mean of cell lines 1–5 with SEM plotted as error bars in experimental passages. Cells were cultivated in the control (control) and experimental media 1–3 (E1–3). Statistical analysis was performed using Friedman’s test followed by Dunn’s post hoc test; ** *p* < 0.005. Individual graphs for each cell line are enclosed in Appendix A.

**Figure 5 biomolecules-11-00022-f005:**
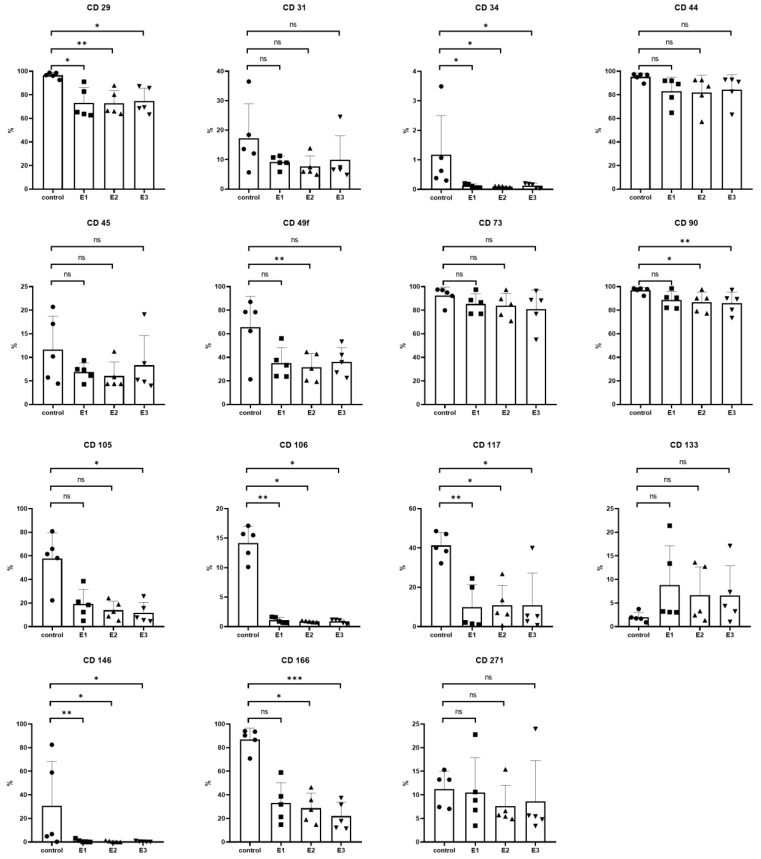
Flow cytometry analysis in the 6th passage of the experiment. The graphs demonstrate the percentages of cells stained by fluorochrome-labeled antibodies against particular cell surface markers in the population. The group mean is plotted as a column, SD is plotted as an error bar, and individual values are plotted as symbols (circle, triangle, and cube). Cells were cultivated in the control (control) and experimental media 1–3 (E1–3). Statistical analysis was performed using Friedman’s test followed by Dunn’s post hoc test; * *p* < 0.05, ** *p* < 0.005, *** *p* < 0.001.

**Table 1 biomolecules-11-00022-t001:** Sex and age of the tooth donors and tooth specification.

Cell Line	Sex	Age	Tooth Specification
Line 1	Female	14	Lower third molar
Line 2	Female	15	Lower third molar
Line 3	Male	13	Upper second premolar
Line 4	Female	18	Lower third molar
Line 5	Female	15	Lower third molar

## Data Availability

Data is contained within the article or Appendix A.

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
