# Peer review of "Low Molecular Weight Hyaluronic Acid Effect on Dental Pulp Stem Cells In Vitro"

_biomolecules, 2020, doi:10.3390/biom11010022_

Round 1

Reviewer 1 Report

The authors shows the potential of three hyaluronic acid fractions for cell therapy using dental pulp stem cells. The experiments do not add newly relevant data to previous manuscripts of the field.

In general, the results shows isolated experiments and a graph of pooled data in order to give an idea of the variation of error. This is a good way to show the degree of effect and reproducibility of the technique. However, this type of individualized data should be moved to supplementary data and removed from the main figures as the data is presented two times in the main figure. The graphs of the figure 1 should be reformatted in order to label properly the Y axis and replace “%” by “Percentage of cell viability”.

It is not clear the rationale of why the authors decide to do the experiment of cell size  and flow cytometry label specifically at 6th passage. For the figure 2 it lacks the pool of the different cell lines to clearly show if there is an effect in the experimental media and not merely for the cell line.

On figure 3 population doubling, are generally maintained but seem to decrease between passage 4 and 6 for the control situation. This is not normal for DPSCs such there are numerous papers showing the potential of these stem cells growing beyond that limit and specially maintaining the CD133 positivity, the author should made a comment trying to explain that.

There is not rationale to split figure 5 into a and b. The authors should merge both in a single figure. Furthermore, the authors should explain why the CD31 (endothelial cell marker) remains stable without any significant change and the marker CD146 (perycite) drops sharply with the experimental media.

Table 2 should be removed, do not add any valuable information as it is redundant with the data presented on figure 5.

Reviewer 2 Report

The manuscript is describing the effect of low MW hyaluronic acid on DPSCs and interesting topic in this area. However the soundness of scientific finding should be improved for the publication to the journal Biomolecules. See below.

Clearer and more precise presentation of the data is required.

For example,

Most figures are not showing reasonable error bars, in particular the results of Fig 5 a and b is unclear because the dimension of error in control group is too high (e.g. in the case of CD 146, there is mostly no difference between control and E1/2/3 groups in 3 of 5 cell lines.
Table 2 will be replaced with graph figure as the readers understand better.

Round 2

Reviewer 2 Report

The authors revised as the reviewer requested.